# Fever-like temperature bursts promote competence development via an HtrA-dependent pathway in *Streptococcus pneumoniae*

**Mickaël Maziero** [1,2], **David Lane** [1,2], **Patrice Polard** [1,2]*, **Mathieu Bergé** [1,2]*

**1** Laboratoire de Microbiologie et Génétique Moléculaires (LMGM), UMR5100, Centre de Biologie Intégrative (CBI), Centre Nationale de la Recherche Scientifique (CNRS), Toulouse, France, **2** Université Paul Sabatier (Toulouse III), Toulouse, France

* mathieu.berge@univ-tlse3.fr (M.B); patrice.polard@univ-tlse3.fr (P.P)

**Data Availability Statement:** All relevant data are within the paper and its Supporting Information files.

## Abstract

*Streptococcus pneumoniae* (the pneumococcus) is well known for its ability to develop competence for natural DNA transformation. Competence development is regulated by an auto-catalytic loop driven by variations in the basal level of transcription of the *comCDE* and *comAB* operons. These genes are part of the early gene regulon that controls expression of the late competence genes known to encode the apparatus of transformation. Several stressful conditions are known to promote competence development, although the induction pathways are remain poorly understood. Here we demonstrate that transient temperature elevation induces an immediate increase in the basal expression level of the *comCDE* operon and early genes that, in turn, stimulates its full induction, including that of the late competence regulon. This thermal regulation depends on the HtrA chaperone/protease and its proteolytic activity. We find that other competence induction stimulus, like norfloxacin, is not conveyed by the HtrA-dependent pathway. This finding strongly suggests that competence can be induced by at least two independent pathways and thus reinforces the view that competence is a general stress response system in the pneumococcus.

## Author summary

*Streptococcus pneumoniae* is a commensal bacterium and an opportunistic pathogen of humans. Certain environmental stimuli, such as a variety of antibiotics targeting distinct cellular functions, trigger the induction of the distinct physiological state of competence, in which cells can, among other things, import and integrate external DNA. Competence is thus considered a general stress response in this highly adaptable species. To understand the role of competence in pneumococcal interaction with its host and in pathogenicity, we have attempted to decipher the pathways that enable appropriate reactions to environmental stress, and have focused here on induction of competence by elevation of temperature to levels similar to that of a host in fever. We found that elevated temperature raises

**Funding:** This work was funded the Agence Nationale de la Recherche (grants ANR-17-CE13-0031 to P.P.). M. M. was supported by an MESR (Ministère de l'Enseignement Supérieur et de la Recherche) fellowship. The funders had no role in study design, data collection and analysis, decision to publish, or preparation of the manuscript.

**Competing interests:** The authors have declared that no competing interests exist.

the basal expression level of the competence control operon, and thus lowers the threshold of transition to full competence induction. By genetic characterisation of the thermal induction pathway of competence, we demonstrated that the chaperone/protease HtrA is essential for relaying of the thermal signal but is not involved in transmitting other stimuli such as those arising from the presence of certain antibiotics. Our work supports the view that competence can be induced through various pathways in response to distinct stimuli, including fever-like bursts of temperature that the pneumococcus could face in its natural habitat.

## Introduction

The human pathogen *Streptococcus pneumoniae* (the pneumococcus) is an opportunistic human pathogen implicated in various diseases such as otitis media, sinusitis, meningitis, pneumonia and septicemia. Methods enlisted to combat the bacterium, such as vaccination and antibiotic treatment, are compromised by the rapid generation of variants resulting from various genome modification events generated in particular by natural transformation. This horizontal gene transfer mechanism occurs only when *S. pneumoniae* has switched to a particular physiological state called competence, coordinated at the population scale by a secreted signaling peptide, termed Competence Stimulating Peptide; (CSP) [1].

Competence development results in the successive transcriptional induction of three regulons, commonly referred to as early, late and delayed competence genes [2–4]. The early competence regulon is controlled by 5 genes organized in two operons, *comAB* and *comCDE*. Together, their expression sets up a positive feedback loop, which is the central regulatory nexus of pneumococcal competence. It begins by production of a pre-peptide from the first gene of the *comCDE* operon that is processed and exported by a dedicated ABC transporter, ComAB, to become the mature 17-residue CSP. The exported CSP [1] activates its receptor ComD, the histidine kinase of the two-component system ComDE [5]. Activation of ComD leads to autophosphorylation, then transmission of the phosphate to its cognate response regulator ComE. Phosphorylated ComE (ComE~P) specifically activates transcription of the early genes. Among them are *comX1* and *comX2*, both encoding an identical alternative sigma factor, $\sigma^x$, and *comW*, encoding its co-factor. Together, they promote transcription of about 60 late competence genes [3,6,7]. Late competence proteins have been implicated in various processes such as natural transformation, fratricide and virulence [8–10]. One of them is DprA, shown below to be relevant to the present work. DprA plays a dual role: it both loads RecA onto transforming DNA to enable recombination [11], and inhibits ComE~P to shut off competence [12,13]. Finally, in a cascading system, delayed genes are induced by unknown molecular mechanisms [2,3].

Only a small set (about 20) of the genes induced during competence are necessary for transformation [3,14]. Why transcription of the other genes is induced remains to be discovered. This observation, together with the ability of several antibiotics and drugs to promote competence development [15–17] and the lack of an SOS regulon in *S. pneumoniae*, led to the suggestion that competence could be considered as a general response to stress in this species [14,16]. If this were so, the competence regulation network would be expected to connect with several molecular pathways involved in stress response.

The vast majority of bacteria encounter physical and chemical changes in their environment that could be sensed as stress. An obvious one is temperature. Temperature variation can affect commensal and opportunistic bacteria such as *S. pneumoniae* that reside in the human

nasopharynx in biofilms and can colonize and infect various organs. Consequently, it is generally accepted that *S. pneumoniae* can encounter strong variations in its direct environment [18], among them abrupt temperature variations between 28 and 37˚C, and even higher during fever [19]. This kind of variation could induce cell responses including production of Heat shock Proteins (HSPs). The *S. pneumoniae* genome encodes various HSPs. Part of the regulation of their synthesis appears similar to that in *B. subtilis*. Indeed, it has been shown that in both *S. pneumoniae* and *B. subtilis* that the well conserved negative regulator CtsR controls expression of *clpP*, *clpE*, *clpC* and *groESL* HSPs [20]. A second regulator previously identified in *B. subtilis*, HrcA, controls two HSP operons, *hrcA-grpE-dnaK-dnaJ* and *groESL* in *S. pneumoniae* [21]. However, our understanding of the regulation of heat response in *S. pneumoniae* remains fragmented as very few global approaches, such as that of Lee et. al (2006), have been adopted [22]. As those proteins known to be involved in temperature adaptation, e.g. HtrA and HrcA, belong to the CiaRH and competence regulons [2,23,24] suggests cross regulation between stress-response pathways. Furthermore, a wide variety of proteins implicated in various aspects of cell maintenance, including various HSPs, are specifically over-produced during competence [2–4]. This observation and other work [25–27] raise the possibility of a link between temperature and competence.

In this work, we demonstrate that Heat shock (HS) induces competence gene expression. We provide genetic evidence that this HS-mediated induction proceeds in two steps: the first, occurring immediately after the HS, increases in a CSP-independent manner the basal transcriptional level of the ComDE regulon, but not that of the ComX regulon; the second depends on CSP and substantially raises transcription of both ComE and ComX regulons. We show that the membrane-bound protease, HtrA, transmits the HS signal that immediate boosts basal transcription of *comCDE*. In all, our results show that bursts of temperature comparable to that of the host fever state promote competence development in *S. pneumoniae* via a new specific HtrA-dependent pathway.

## Results

### Heat Shock (HS) promotes pneumococcal competence development

To test the effect of temperature variations on competence development, we first monitored transcription of the early competence gene *comC* in exponentially growing cells heated for various times at various temperatures. We used a strain carrying a fusion of the firefly luciferase (*luc*) gene to *comC*, and recorded luciferase activity by luminometry as a proxy for competence development [15,28]. To minimize the possibility of indirect effects of transient temperature changes on medium and growth, we sought to determine the minimal temperature and exposure time necessary to influence the development of competence.

Cells were incubated at 37˚C in medium with initial pH adjusted to 7.0 so that cells do not induce competence spontaneously [15]. Cells were then exposed to various temperatures for 15 minutes, before measurement of competence development at 37˚C (see materials and methods). Fig 1A (left panel) indicates that cells transiently exposed to higher temperatures developed competence 80 to 200 minutes after return to 37˚C, in contrast to cells continuously maintained at 37˚C. Competence developed earlier as exposure temperatures increased up to 42˚C. In contrast, at 45˚C, competence induction appeared both delayed and of lower amplitude, owing to the prolonged pause or cell death prior to resumption of growth (Fig 1A, right panel). As 42˚C appeared to be optimal for promoting competence, we then tested the ability of cells to develop competence after exposure to 42˚C for various periods of time (Fig 1B, left panel). Our results indicate that short exposure times at this temperature (5 minutes) are sufficient to promote competence development. In addition, we found that the onset of competence was triggered earlier with increasing heat exposure times up to 15 minutes, as observed for biochemical stress

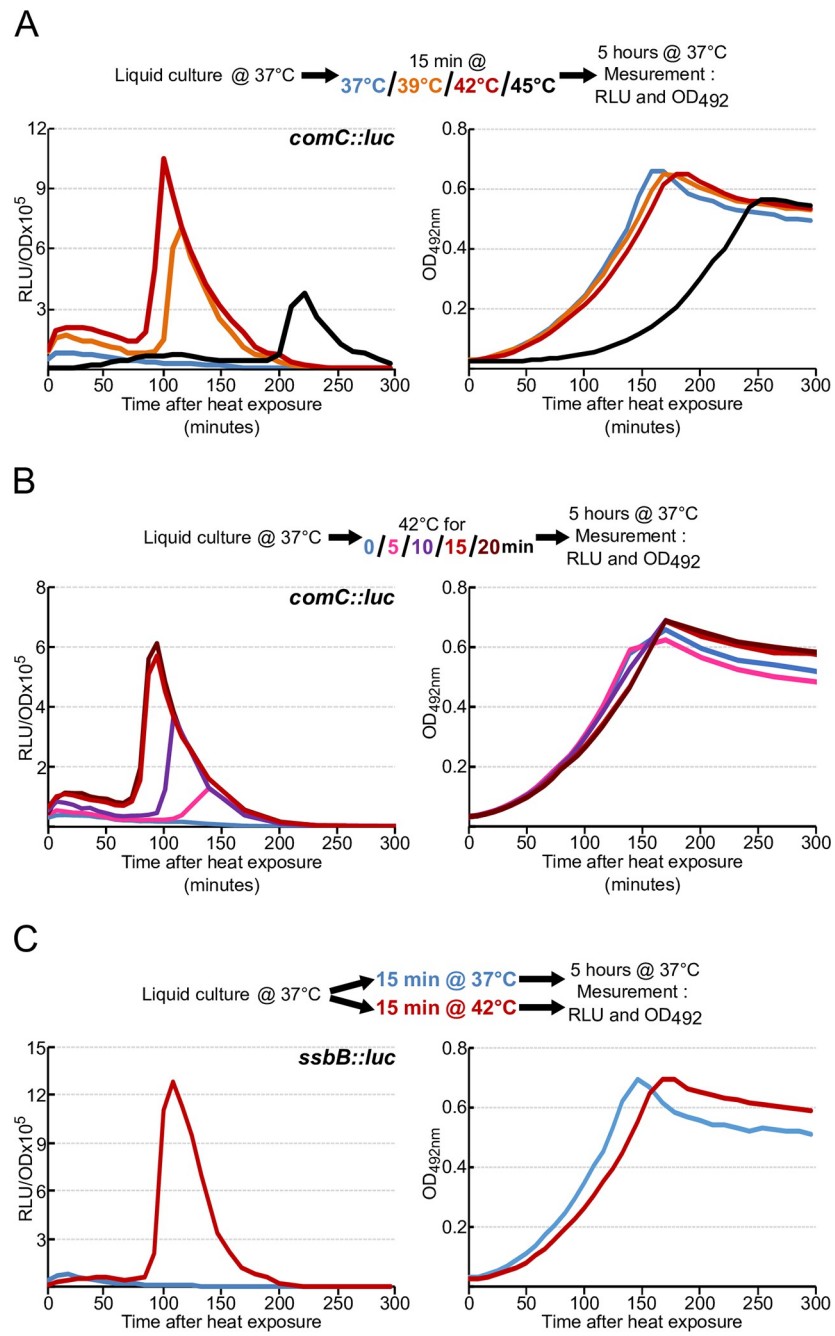

**Fig 1. Heat exposure promotes competence development.** A. *comCDE* expression was monitored in strain R825 growing in C+Y medium at 37°C after 15 minutes of exposure to 37, 39, 42 or 45°C, as shown in the colour key, prior to the first measurement at time 0. Values are expressed in relative light units per OD (RLU/OD) in the left panel and the corresponding growth curves reported as $OD_{492nm}$ on the right. B. As in A, but for various periods at 42°C only. C. As in A, but for *ssbB*::*luc* in strain R895 after 15 minutes at 42°C only. For clarity, only a single data set, representative of at least three independent determinations made on different days, is presented.

[29,30]. Importantly, similar results were obtained with a strain containing a transcriptional fusion of the *luc* gene with the promoter of a late-competence gene, *ssbB*, when this strain was exposed to 42°C for 15 minutes (Fig 1C). To check that these competence triggers were

biologically effective, we evaluated natural transformation under the different conditions, and found that only heat-exposed cells showed a significant rate of transformation (S1A–S1C Fig). We conclude that transient exposure to 42˚C stimulates induction of both early and late competence genes in a time, and dose dependent manner.

## HS mediates an immediate and CSP-independent burst of *comCDE* expression

Our data indicate that *comCDE* expression rises massively 75 minutes after 15 minutes at 42˚C, as manifested by the large peak of luciferase activity that extends for more than 100 minutes in cells containing the *comC::luc* fusion (Figs 1A–1C and 2A). Interestingly, we also detected a significant smaller burst of luciferase activity right after the HS which was maintained until full induction. The amplitude of this first burst of *comC::luc* expression varied with the intensity and duration of the HS (Figs 1A and 1B and 2A). In contrast, the *ssbB::luc* fusion strain, which reports late competence gene expression, did not generate the immediate small burst of luciferase activity right after the HS, but only the large peak 75 minutes later (Fig 1C). This observation suggested that HS induced a small immediate increase in the basal level of *comCDE* expression, which then promotes a strong subsequent induction of early and late competence genes through CSP export. To test this hypothesis, we repeated heat shock experiment with a *comA⁻* strain, which is unable to export CSP (Fig 2A, enlarged in Fig 2B). This strain was unable to induce the wholesale peak of competence gene expression, even though the initial burst of *comC::luc* expression just after HS was still present. This also indicates that HS immediately and transiently increases *comCDE* expression independently of an extra-cellular interaction between CSP and ComD. Nevertheless, the decrease in expression observed in a *comA⁻* mutant compared to that in the wild-type strain suggests that CSP is needed to maintain this overproduction and to trigger efficient competence induction further down the line. The addition of a saturating amount of CSP 20 minutes after HS resulted in the immediate triggering of the autocatalytic loop, suggesting that the available CSP was indeed a limiting factor (S2A and S2B Fig). It should be noted that these results were also obtained with a *comC⁻* strain, incapable of producing CSP (strain *comC0*) (Fig 3B). Subsequently, the *comC0* and *comA⁻* strains were used interchangeably to characterize the initial burst of *comCDE* transcription.

The first HS-induced burst of *comCDE* expression could be the result of a small increase in all cells of the population or to a strong increase in a subpopulation. To distinguish these possibilities, we used fluorescence microscopy to compare expression of the *gfp* gene from the *comCDE* promoter ($P_E$) in cells transiently exposed or not to HS at 42˚C. As previously observed with the same $P_E$-*gfp* transcriptional fusion at the native *comCDE* locus [31], we detected a very heterogenous basal level of GFP synthesis amongst cells not exposed to HS (Fig 2C, top panels). Notably, the GFP signal appeared brighter in most of the cells right after HS (Fig 2C, bottom panels). Quantification of mean fluorescence per cell further revealed a significant increase of GFP synthesis upon HS in all cells. This higher level of GFP, like observed before the HS, was still heterogeneous. (Figs 2D and 2E). We conclude that while HS immediately induced *comCDE* expression in most cells, the data do not rule out a modification of the degree of heterogeneity in individual cell response.

## HS promotes overexpression of early competence genes

Two promoters are known to drive *comCDE* basal transcription [28]: the competence promoter controlled by ComE, $P_E$, and the promoter, $P_t$, of the gene encoding the *tRNA^{arg5}*, located just upstream of the *comCDE* operon (Figs 3A, S3A). To determine whether $P_t$ is involved in the immediate HS-mediated induction burst of *comCDE* expression, we used a

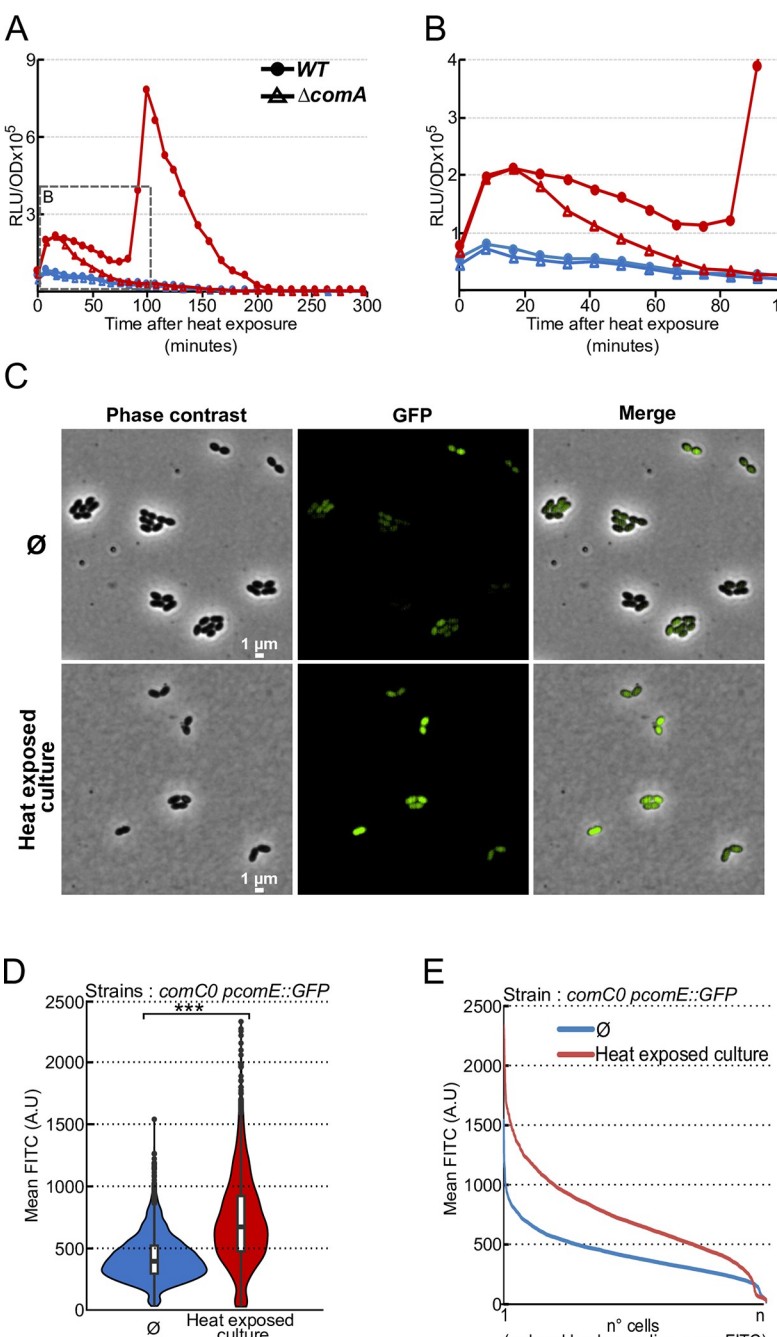

**Fig 2. Heat-induced *comCDE* expression is independent of CSP.** A. As in Fig 1 for strain R825 (solid symbols) and the *comA* strain, R875 (open symbols) exposed for 15 minutes to 42˚C (red triangles) or 37˚C (blue circles). For clarity, only a single data set, representative of three independent determinations made on different days, is presented. B. Enlargement of the early post-heat growth phase from panel A. C. Fluorescence in cells of strain R4254 carrying the *gfp* gene under the control of a ComE-dependent promoter 20 min after return to 37˚C of cells not exposed (top) or exposed (down) to 42˚C for 15 minutes. D. Violin plots representing mean GFP fluorescence intensity in non-HS (blue) and HS (red) cells. Boxes extend from the 25th percentile to the 75th percentile, with the horizontal line at the median. Dots represent outliers. n = 3161 cells analysed for non-HS conditions; n = 2545 cells analysed for HS conditions. *** = p-value < 0.001. E. Alternative representation of the data in D, as mean GFP fluorescence of each cell ordered by decreasing mean fluorescence intensity. Blue: non-heat-exposed cells; red: heat-exposed cells.

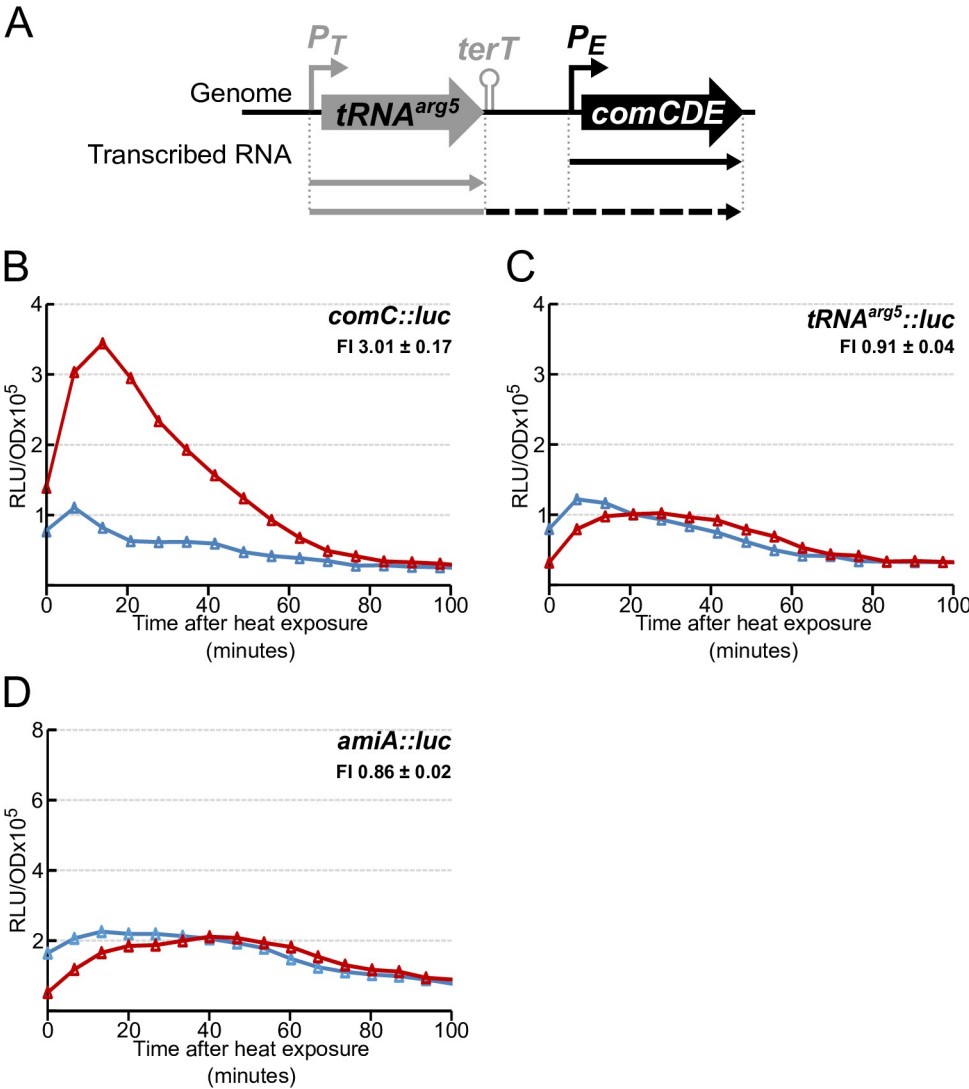

**Fig 3. Heat-induced *comCDE* expression is specific to the $P_E$ class promoter.** A. Schematic representation of the *comCDE* locus and upstream region. Promoters driving the basal level of *comCDE* transcription are represented by bent arrows. Straight arrows represent transcription activity; the dashed arrow represents transcription bypassing the terminator terT. B-C. Gene expression measured as in Figs 1 and 2 following 15 minutes exposure to 42°C of strains R1521 (*comC::luc*), R1694 (*tRNA^arg5^::luc*) and R4639 (*amiA::luc*). FI = Fold induction of gene expression in HS condition compared to non-HS. This number under the genotype indicates the mean of the estimated induction ratio followed by the corresponding standard deviation for at least 3 independent experiments (see materials and methods). Corresponding growth curves are presented in S3 Fig.

$P_{tRNA}{}^{arg5}\_luc$ transcriptional fusion at the native locus [28] to monitor luciferase activity of untreated and HS-treated cells. No HS-mediated increase of transcription was observed with the $P_{tRNA}{}^{arg5}\_luc$ reporter, in contrast to the $P_E$-*luc* reporter (Figs 3B and 3C, S3B, S3C). The luciferase expression profile from $P_{tRNA}{}^{arg5}\_luc$ was similar to that of a transcriptional fusion of *luc* with the promoter of the *amiA* gene (Figs 3D, S3D), known not to be part of the competence regulon [2,15]. HS even promoted a modest decrease in luciferase expression from $P_{tRNA}{}^{arg5}$and $P_{ami}$ promoters (Fig 3C and 3D), consistent with the known slight thermosensitivity of luciferase activity [23]. These results suggested that the burst of *comCDE* transcription depends solely on the $P_E$ promoter. Since $P_E$ class promoters control transcription of early

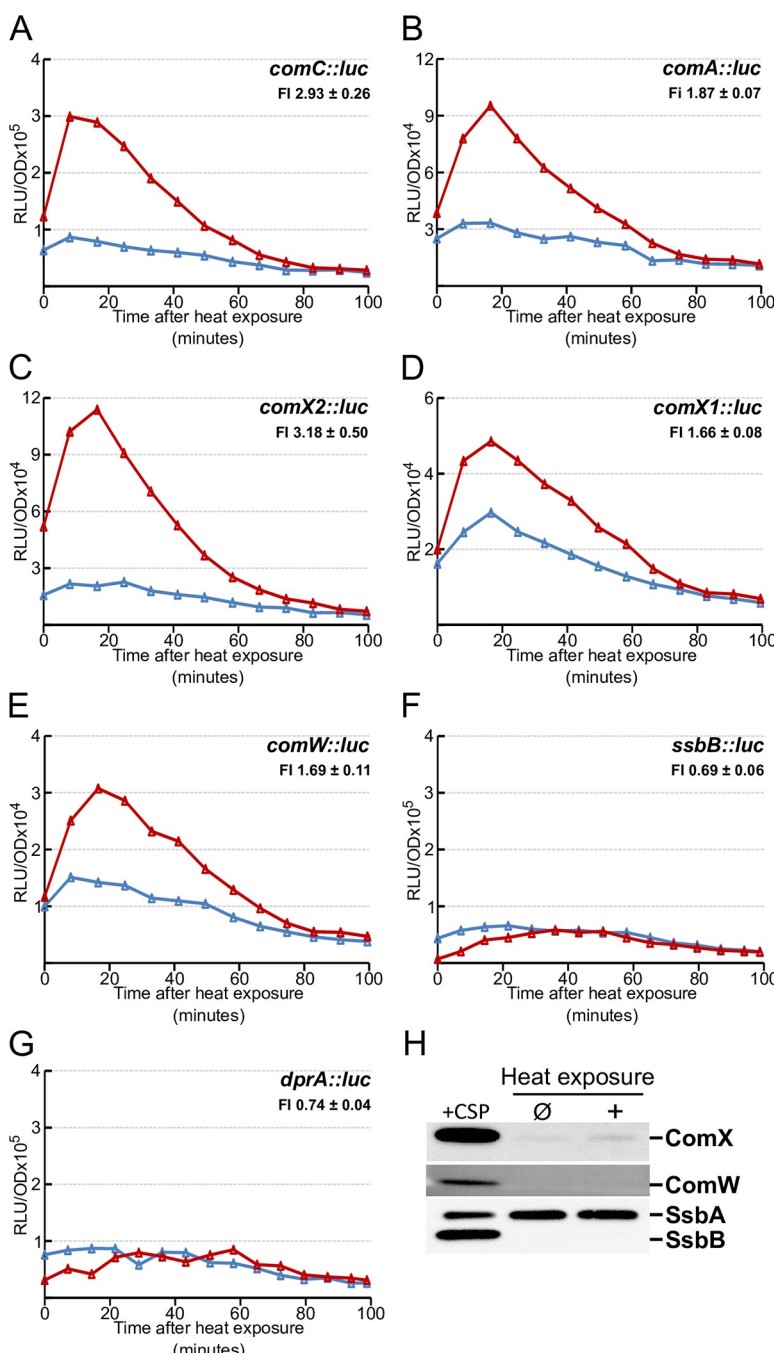

**Fig 4. The burst after heat exposure is specific to early competence genes.** Gene expression measured as in Fig 3 following 15 minutes exposure to 42˚C of strains: A. R1521 (*comC::luc*); B. R1548 (*comA::luc*); C. R2200 (*comX2::luc*); D. R2218 (*comX1*::luc); E. R3688 (*comW::luc*); F. R1502 (*ssbB::luc*), G. R2448 (*dprA::luc*). H. Western blot comparing cellular levels of ComX, ComW, SsbB and SsbA in CSP induced cells, non-heat exposed cells (control) and heat exposed cells. FI = Fold induction of gene expression in HS conditions (see materials and methods). Corresponding growth curves are presented in S4 Fig.

competence genes [15,31], we next examined the effect of HS on another four of them, i.e., those controlling expression of *comA*, *comX1*, *comX2* and *comW*, using transcriptional *luc* fusions. HS induced an immediate increase in their expression (Figs 4B–4E and S4B–S4E), as

with *comC* (Figs 4A and S4A). These results indicate that HS-induced ComE-controlled transcription of early competence genes is general and not restricted to *comCDE*.

As the two identical copies of *comX* (*comX1* and *comX2*) encode the alternative sigma factor, $\sigma^X$, which with the assistance of ComW promotes optimal expression of late competence genes [7,32], we tested whether HS also boosts transcription of two late genes, *ssbB* and *dprA*, in *comC0* cells. The results show that it does not (Figs 4F and 4G, S4F and S4G), corroborating observations previously made in wild-type (*comC*+) cells (Fig 1C).

Attempting to understand why the HS-stimulation of *comX* and *comW* transcription does not raise late competence gene transcription in a *comC0* background, we measured the quantities of $\sigma^X$ and ComW produced after HS relative to those made upon addition of synthetic CSP. As shown in Fig 4H, Western blot analysis reveals that $\sigma^X$ and ComW remain barely detectable 20 minutes after HS, a time corresponding to the maximum of the *comX* and *comW* transcriptional burst (Fig 4C–4E). Even though $\sigma^X$ increased slightly after HS, it remained far below that observed following CSP induction (Fig 4H). In addition, SsbB, the most abundant late protein [32], remains undetectable after HS whereas the quantity of the constitutively expressed SsbA protein [32], used as internal control, remains constant (Fig 4H). We concluded that the level of expression of early competence genes can be decoupled from that of late genes in *S. pneumoniae*.

## The immediate HS-mediated *comCDE* expression burst depends on the ComDE two component system

Our results (Figs 3 and 4) had shown that the immediate HS-mediated *comCDE* transcriptional burst strictly depends on $P_E$. As basal $P_E$ activity is controlled by the ComDE two component system [28,33,34], we explored whether ComD or ComE activity is necessary for the HS-induced burst in *comCDE* expression (Figs 5 and S5). We monitored luciferase activity in *comC::luc* strains with null mutations in *comD* or *comE*. Inactivation of *comE* reduces *comCDE* basal expression, as previously observed [28], and also abolishes the HS burst of transcription observed in a *comC*- background (Figs 5A and 5B, S5A, S5B). We observed the same pattern of *comCDE* expression using a ComE$^{D58A}$ point mutant, in which phosphorylation of the transcriptional activator ComE is abolished (Figs 5C and 5D, S5C, S5D). As ComE phosphorylation is presumably performed by its cognate histidine kinase ComD [13,28,31], we examined *comCDE* transcription after HS in a *comD*- strain. Lack of ComD abolished the HS-induced expression burst (Figs 5E and 5F, S5E, S5F). This strongly suggests that phosphorylation of ComE by ComD is needed to transmit HS signal.

## The immediate HS-induced burst in *comCDE* transcription depends on the HtrA protease

To identify the factors that modulate *comCDE* expression after HS, we monitored HS-induced expression of the *comC::luc* fusion in three *comC0* strains mutated for a specific candidate gene. The selection of the candidates was based on their involvement in *comCDE* regulation or their association with HS, or both. The first tested was *dprA*, as it is the main regulator of ComE owing to its ability to inhibit phosphorylated ComE transcriptional activity [12,35]. The second was c*lpP*, encoding a heat shock protein previously described as a negative regulator of *comCDE* expression [20]. The third was *htrA* (High Temperature Requirement A), encoding a protein described in *E. coli* as required for growth at elevated temperature [36,37] and also known to affect competence development or natural transformation in *S. pneumoniae* [38–41].

As shown in Fig 6B and 6C, *dprA* and *clpP* mutations do not abolish the increase *comCDE* expression right after HS. In contrast, the *htrA* mutant showed no such HS-mediated burst of

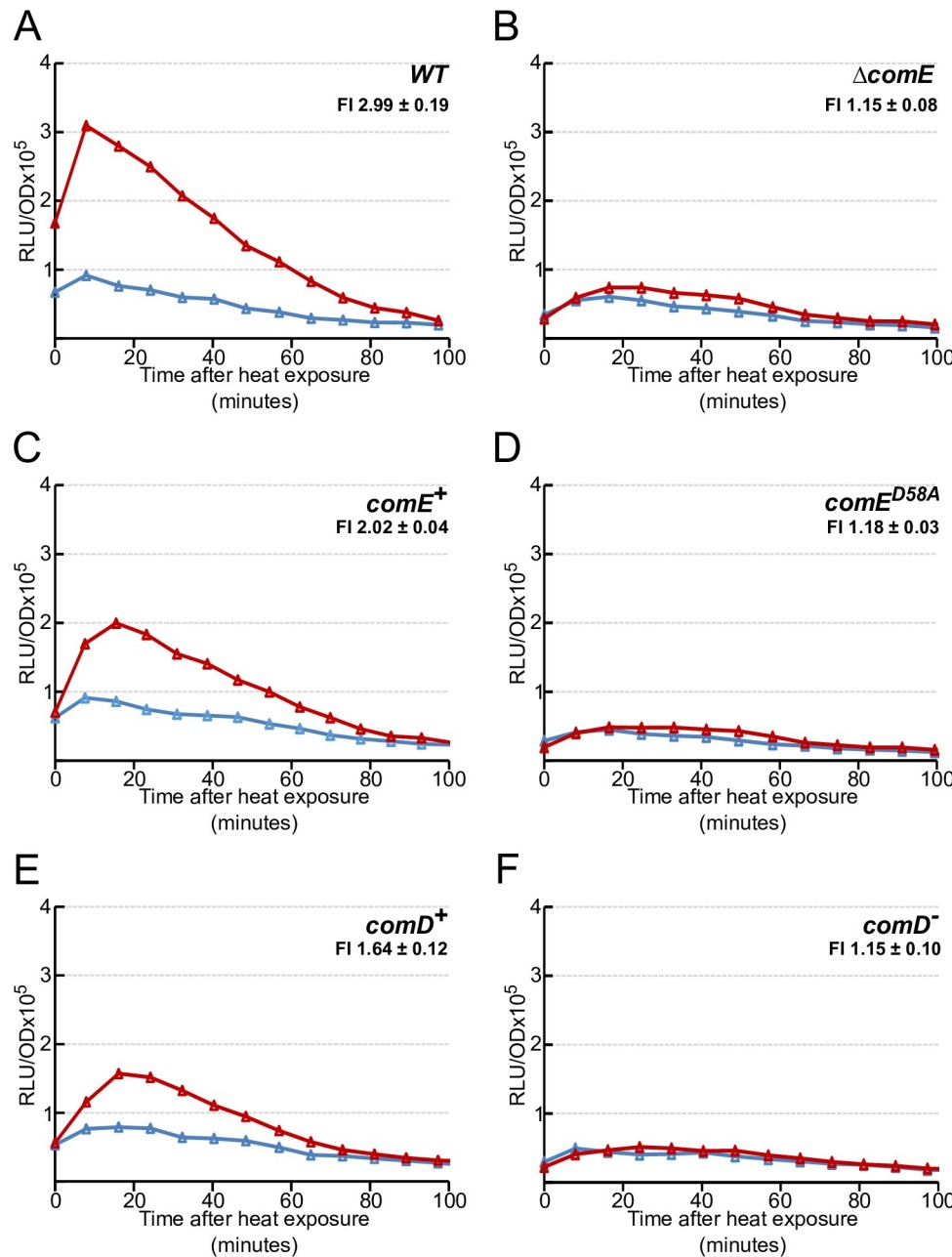

**Fig 5. The *comCDE* burst requires a functional ComD/E two component system.** Gene expression measured as in Fig 3 following 15 minutes exposure to 42˚C of strains: A. R1521 (*comC::luc*); B. R1627 (*comC::luc, comE⁻*); C. 4585 (*comC::luc, hexA⁻*); D. R1798 (*comC::luc, hexA⁻, comE^{D58A}*); E. R1628 (*comC::luc, comD⁺*), F. R1648 (*comC::luc, comD⁻*). FI = Fold induction of gene expression in HS conditions (see materials and methods). Corresponding growth curves are presented in S5 Fig.

*comCDE* transcription (Figs 6D, S6D). This absence of the *comCDE* expression burst was also observed in a *comC⁺/htrA⁻* genetic background. As expected, no triggering of competence was observed 75 minutes later, nor was there any significant level of transformation (Fig 6E–6H). We also confirmed that an *htrA* strain carrying the PE-*gfp* transcriptional fusion is unable to produce an increase of GFP fluorescence intensity comparable to that obtained in the wild-

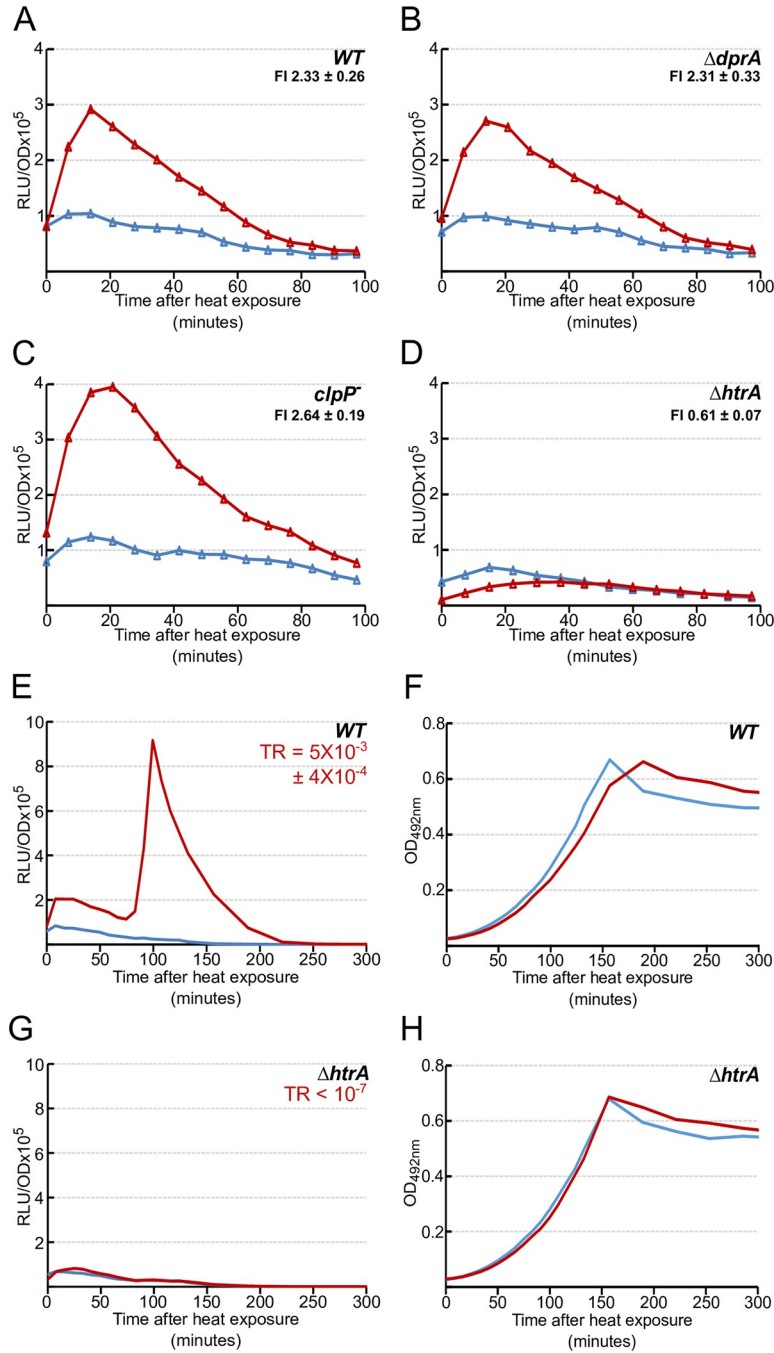

**Fig 6. HtrA is required to conduct the heat exposure signal to *comCDE*.** Gene expression measured as in Fig 3 following 15 minutes exposure to 42˚C of strains: A. R1521 (*comC::luc*); B. Strain R2017 (*comC::luc, dprA⁻*); C. R1526 (*comC::luc, clpP⁻*); D. Strain R4578 (*comC::luc, htrA⁻*). FI = Fold induction of gene expression in HS conditions (see materials and methods). Corresponding growth curves are presented in S6 Fig. E-. *comCDE* expression was monitored in strain R825 (*comC⁺* genetic background) growing in C+Y medium at 37˚C after 15 minutes of exposure to 42˚C (red) or not (blue) prior to the first measurement at time 0. F. Growth curves of the cultures in E. G-H. as in E-F, but in strain R1262 (*htrA⁻*, *comC⁺* genetic background). Note that the *htrA⁻* strain does not show significant growth retardation, as growth takes place at 37˚C following transient exposure at high temperature. TR = Transformation rate (see materials and methods). Data represented as Mean ± standard deviation of triplicate repeats.

type strain in heat-exposed cultures (S6E and S6F Fig). These observations strongly suggest that HtrA is required to promote increase of *comCDE* basal level in response to HS. However, *htrA* forms an operon with the downstream *parB* gene (Figs 7A, S7A). ParB has been shown to influence competence [42], raising the possibility that the HS-mediated response of the *htrA⁻* strain on *comCDE* expression could result from a polar effect on ParB production. To test this possibility, we examined the effect of HS in a *parB* mutant carrying the *comC::luc* transcriptional fusion (Figs 7C, S7C). *comCDE* expression still increased after HS in this strain. We also tested two *htrA* mutants unlikely to impart polar effects on *parB*: one with a stop codon and frameshift near the 5' end, and the other a mis-sense mutation (S234A) that inactivates the canonical HtrA proteolytic active site [43] (Figs 7D, 7E, S7D, S7E). These two strains display a profile of *comCDE* expression after HS similar to that of the *htrA* null mutant (Fig 6D). These results strongly suggest that HtrA and its catalytic site are required to convert the HS stress signal to an increase in basal *comCDE* transcription.

We then explored the possibility that HtrA could be allosterically activated by HS, like its *Escherichia coli* paralogues DegP and DegS. Allosteric activation of DegS depends on loss of interaction between the "L3 loop" of the catalytic domain and the PDZ domain of the protein [44]. Two key residues necessary for this activation in *E. coli* DegS and DegP are conserved in *S. pneumoniae* HtrA (R206 and Q224). As shown in Fig 7F and 7G, mutation of either of them eliminates HS-induced *comCDE* expression. HS had no significant effect on expression of an *htrA:luc* fusion gene (Figs 7H, and S7H), implying that the transient heat stress probably does not alter HtrA protein levels. These results suggest that HS could allosterically stimulate catalytic activity of HtrA, which is necessary to relay the HS signal to *comCDE* transcription.

## Brief antibiotic exposure also induces immediate *comCDE* transcription

Certain antibiotics are known to induce *S. pneumoniae* competence, namely those that damage DNA (e.g. norfloxacin) and those that target ribosomes (e.g. streptomycin) [15]. The effect of others has yet to be reported. However, this induction was observed in cells subjected to continuous antibiotic exposure. We tested whether transient exposure to the same concentrations of norfloxacin or streptomycin, for the same times as HS, also induces a *comCDE* burst. To this end, we monitored *comC::luc* expression after 15 minutes of treatment with each antibiotic, followed by transfer to antibiotic-free medium. Norfloxacin (Figs 8A, S8A, S8E upper panels) exposure promoted a burst of *comCDE* expression. It is important to note that this induction is CSP-independent since it was observed in a *comC0* background. We then asked whether HtrA is required to transmit the induction signal generated by norfloxacin. *comCDE* overexpression was still seen in an *htrA* strain treated with norfloxacin (Figs 8B, S8B, S8E lower panels). Hence, this burst of induction is HtrA independent. Moreover, the observation that transcription from the $P_{tRNA}^{arg5}$-*luc* promoter is also increased (Figs 8C, S8C) while transcription from the *amiA* control gene is not (Figs 8D, S8D) supports the idea that the signalling pathway relaying norfloxacin stress is different from that of heat stress.

Streptomycin also induces a reproducible burst of *comCDE* transcription 40 minutes after antibiotic exposure (S9A Fig upper panels), while no burst was detected for the *amiA::luc* control reporter S9B Fig. This burst is slightly weaker in the *htrA* strain (S9A Fig lower panels), indicating that HtrA might transmit at least some of the stress generated by streptomycin.

## Discussion

### Heat shock stimulates competence development

Here we report a previously unseen causal link between HS and competence induction in *S. pneumoniae*. It is particularly interesting that a simple transient two-degree shift from 37 to

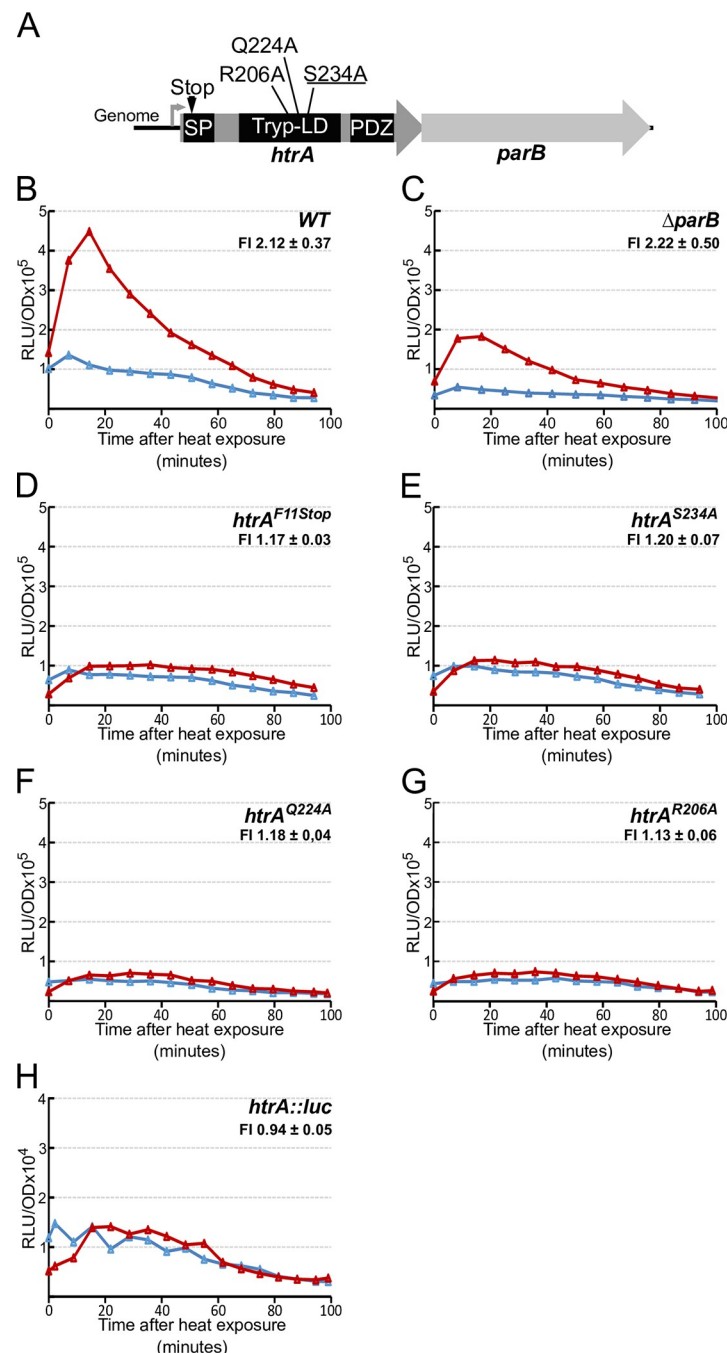

**Fig 7. Catalytic and activation residues of HtrA protease are required to activate *comCDE* transcription.** A. Schematic representation of the *htrA/parB* operon and position of relevant residues of HtrA. B-H. Gene expression measured as in Fig 3 following 15 minutes exposure to 42°C of strains: B. R875 (*comC::luc*); C. R4657 (*comC::luc*, *parB⁻*); D. R4629 (*comC::luc*, *htrA^stop*); E. R4630 (*comC::luc*, *htrA^S234A*); F. R4676 (*comC::luc*, *htrA^Q224A*); G. R4684 (*comC::luc*, *htrA^R206A*); H. *htrA* expression in strain R2813 (*htrA::luc*). Note that the lower basal level for strain R4657 is due solely to experimental variability and is not significant. FI = Fold induction of gene expression in HS condition (see materials and methods). Corresponding growth curves are presented in S7 Fig.

39°C, close to host fever temperature, can promote expression of *comCDE* (Fig 1A). Despite this, previous work on HS or growth temperature variation in *S. pneumoniae* had not revealed any relation with competence development [22,45,46]. In these studies, samples were collected

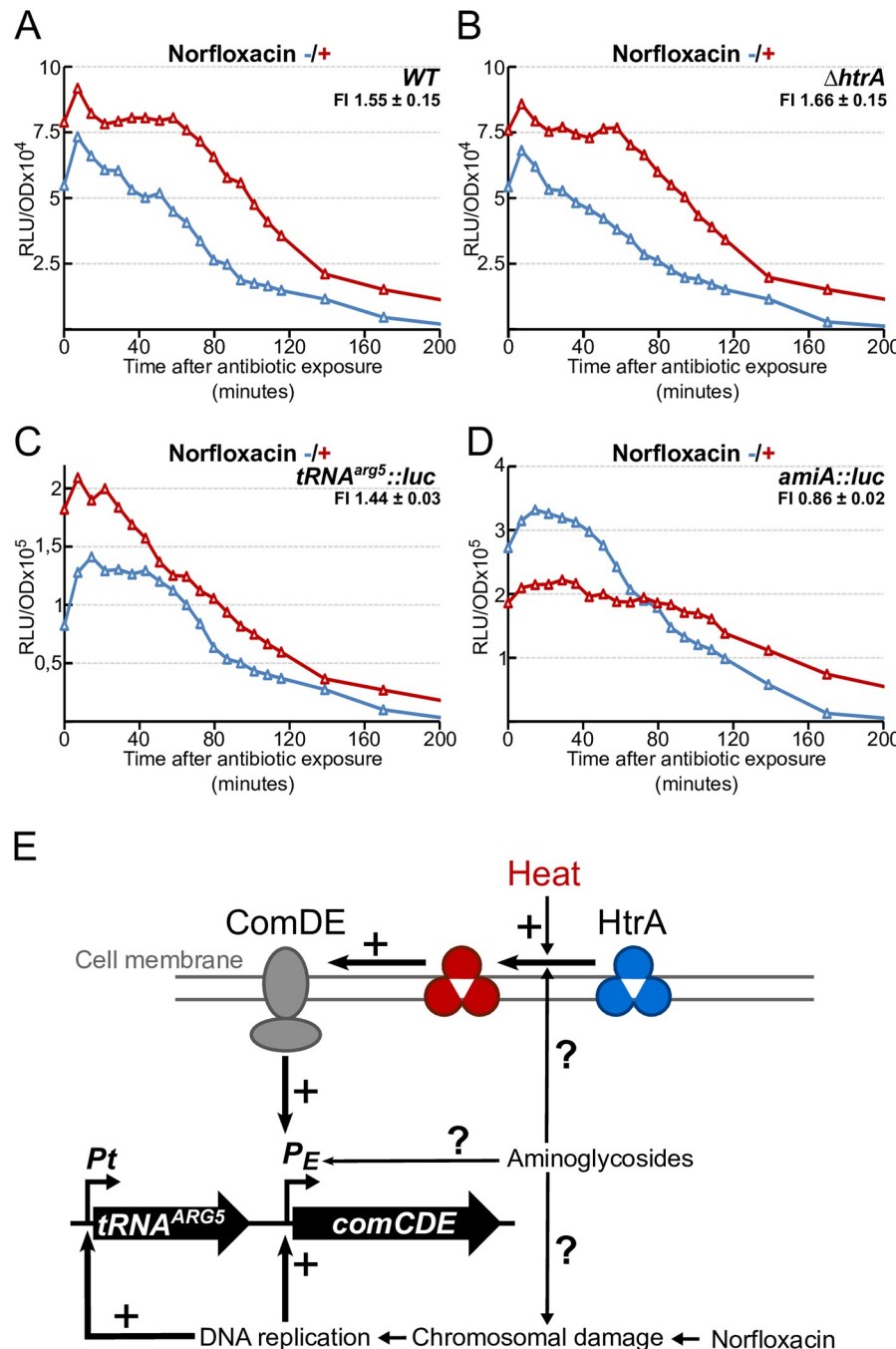

**Fig 8. Several molecular pathways convey different signals to modulate *comCDE* basal level transcription.** *comCDE* expression shown as relative luminescence unit per OD (RLU/OD) in strains growing in C+Y medium at 37°C after 15 minutes exposure to Norfloxacin (1.25 μg mL⁻¹) (red symbols) or no antibiotic (blue symbols) prior to first measurement. A. R875 (parental strain); B. R4629 (*htrA*); C. R4642 (*tRNA*^*arg5*^::*luc*); D. R4641 (*amiA::luc*). FI = Fold induction of gene expression in antibiotic exposure conditions (see materials and methods). Corresponding growth curves are presented in S8 Fig. E. Summary of characterized and potential molecular CSP-independent pathways driving *comCDE* basal expression.

and analysed just after HS or at a specific optical density. However, *S. pneumoniae* competence development is a propagation mechanism initiated by a stressed fraction of the population [15,17,29]. As shown in this study, in HS conditions, the stress induces an initial burst of *comCDE* expression that is maintained in the population via CSP secretion, but real competence synchronization of the entire population occurs at least 75 minutes later. The reasons for this delay remain to be explored but could be sought in the availability of the exported CSP (see below). In addition, the previous studies were not conducted at times or in medium allowing visualization of the impact of HS on competence development in the whole population. Thus, only the most sensitive assays, such as the luciferase-based one used here, would be able to detect the small burst of early competence genes we see, which appears to promote competence development in the whole population.

## Heat shock leads to a specific increase of early gene transcription

We have shown that HS promotes an immediate increase in *comCDE* transcription, and presumably the resulting increase in transcription of the early competence regulon controlled by ComDE. As a result, transcription of early competence genes implicated in competence regulation, like *comA*, *comX* and *comW*, is also increased (Fig 4). Strikingly, late competence gene transcription remains unchanged before competence propagation through the population by the action of CSP. The decoupling of early and late gene transcription is intriguing as induction of early competence genes, $comX_1$, $comX_2$ and *comW* generally induces late gene expression [47,48]. However, in all previous studies, ComX ($\sigma^x$) and ComW were produced at high levels to mimic wholesale competence induction. As ComX and ComW compete with the primary sigma factor, $\sigma^A$ (encoded by *rpoD*) for access to RNA polymerase [25,49,50], a high ComX-ComW/$\sigma^A$ ratio could be essential to late gene transcription. As a result, it is possible that soon after HS, cellular ComW and ComX remain below the levels required to displace $\sigma^A$ and induce the late competence genes.

Whatever the underlying mechanism, the biological significance of this temporal decoupling may be a key element in allowing a stable increase of *comCDE* transcription. Indeed, as the late protein DprA is the main inhibitor of phosphorylated ComE [12,35], an unduly tight coupling between early and late genes could generate a negative feedback loop on early gene expression. In this way, the lack of DprA production allows a lasting increase in expression of early genes, especially of the *comCDE* operon, in the whole population. This kind of regulation could generate a more alert population prone to trigger competence when conditions become optimal.

## HtrA converts heat exposure into increased basal *comCDE* transcription

Several reports have described a relationship between HtrA and competence for genetic transformation [38–41,43]. Stevens et al. propose that HtrA conveys ribosome decoding error signals to competence development [41]. The same group later showed that HtrA is able to degrade CSP *in vitro* [38]. They proposed that HtrA inhibits competence by constitutively degrading extracellular CSP, but that at high levels of decoding error, HtrA is monopolised by incorrectly folded proteins, allowing an increase of extracellular CSP concentration and full competence development [38,41].

In this work, we show that HtrA converts HS stress into an increase in the basal transcription of *comCDE*. Two of our observations are incompatible with Stevens et al's proposal for the role of HtrA in relaying HS; (i) HS induction is CSP independent (Fig 2A and 2B); (ii) basal *comCDE* expression did not increase in an *htrA* mutant (Fig 6D). These observations lead us to propose that HtrA could also act as an activator during transient heat variation.

## Conserved allosteric activation residues of HtrA are necessary for transduction of the HS signal to competence induction

HtrA family proteins display chaperone and protease activities, and switching between them can be temperature dependent [51]. Although the duality of the function remains debatable, protease activity seems to predominate [52]. HtrA-like proteins possess a trypsin-like protease domain and one PDZ domain, as in *S. pneumoniae*, or more. Among prokaryotes, DegS and DegP of *E. coli* are the best structurally characterized paralogues of this family. DegS protease activity requires remodeling of the protease-domain L3 loop. The PDZ domain of DegS acts as a lock for the active site and requires conformational changes to stabilize the active form [53,54]. Here we demonstrate that mutations known to abolish protease activity prevent the HS signal from increasing *comCDE* transcription (Fig 7E). Moreover, presumed "activation residues" of the L3 loop are required (Fig 7F and 7G). These observations suggest how HtrA-mediated HS activation *in S. pneumoniae* might resemble that seen in *E. coli*. In *E. coli*, heat and environmental stress leads to the accumulation of misfolded proteins that activate DegS. C-terminal peptides (COOH-YYF) of outer membrane proteins are the best characterized activators of DegS [44]. Several DegS residues have previously been shown to bind outer membrane protein peptides, and equivalent HtrA residues have been identified *in vitro* [55]. Ongoing experiments in our laboratory suggest that these residues are not implicated in the HS-dependent burst of transcription of *comCDE*. As outer membrane proteins are not present in *S. pneumoniae*, we propose that other misfolded proteins or peptides, presently unknown, could break inhibitory L3 loop/PDZ interactions. Such misfolded proteins are known to be generated during HS and by kanamycin [56] and streptomycin [57] treatment. Alternatively, HtrA could be activated by a PDZ-independent pathway, as is human HtrA1 [58].

## HtrA is an activator of *comCDE* transcription

Because protease activity of HtrA appears necessary for transduction of the HS signal, we envisage at least two mechanisms to explain the increase in basal *comCDE* transcription. First, HtrA could promote degradation of a transcription repressor, as already observed for the HtrA family protein DegS of *E. coli* [44,59,60]. However, the increase in *comCDE* transcription after HS cannot be explained by simple transcriptional derepression as ComD, ComE or ComE-P are themselves still required to ensure the burst of *comCDE* transcription. Second, it is possible that activated HtrA could promote the phosphorylation cascade of the ComDE two component system directly or indirectly by degrading an unknown ComDE repressor. Alternatively, a mixed model could apply, in which removal of a transcriptional repressor allows ComE-P-dependent activation of the $P_E$ promoter.

It is intriguing that despite the initial burst of *comCDE* transcription and the resulting increase of *comC* and *comA* expression, no autocatalytic loop is launched. This suggests that CSP activation of the autocatalytic loop that alerts the whole population was not efficient. Since HtrA can degrade CSP, at least *in vitro* [38], a simple hypothesis is that an activated HtrA on one hand increases *comCDE* transcription but on the other hand degrades the overproduced secreted CSP. Another option is that HtrA could antagonize ComAB function, as previously hypothesized [61]. In either of these ways, a subtle new equilibrium could be reached between induction and repression of CSP/ComDE production, leading to a more vigilant pneumococcal population rather than to immediate promotion of an uncontrolled and massive response. During subsequent growth, additional signals modulate this level of expression until reaching a threshold which triggers competence induction and the irreversible propagation to the whole population. HtrA would therefore have two actions on competence induction, one positive through its ability to increase *comCDE* transcription, and the other

negative through CSP degradation. Depending on the stimulus and physiological conditions analysed, one of these actions would predominate. Thus, in the case of modulation of translation accuracy [41], degradation of the CSP would be the dominant element regulating the onset of competence.

### Several independent pathways relay distinct stress to the *comCDE* operon

Previously, we demonstrated that HS, like ribosomal or chromosomal damage, promotes competence development [15]. In this study we show that, like HS, streptomycin and norfloxacin induce an initial burst of *comCDE* transcription just after stress application (Figs 8A, S8E, S9). The lack of CSP involvement in promoting increased *comCDE* expression is a solid argument for viewing the stress as applying to individual cells and their internal reactions. Above all, the observation that this burst could be HtrA-dependent (Fig 6D) or independent (Figs 8B, S9), suggests that various pathways relay stress to transcription of the *comCDE* operon (Fig 8E). Clearly, at least two individual pathways exist, an HtrA dependent one and at least one independent of HtrA. HtrA conveys all HS signals, but DNA damage generated by norfloxacin, for example, activates *comCDE* transcription via an HtrA-independent pathway. Chromosomal damage has been proposed to raise *comCDE* expression via an increase in its gene dosage [62], or by signalling from arrested replication forks [30] which would be consistent with the observation that the $P_{tRNA}^{arg5}$ promoter is also increased by the action of norfloxacin. Streptomycin stress appears to be conveyed to *comCDE* partially via an HtrA-dependent pathway; indeed, the lower than wild type burst seen in an *htrA* mutant (S9 Fig) implies a contribution from HtrA. This would be consistent with previous work showing that antibiotics that perturb ribosome activity lead to competence development via a pathway implicating HtrA [41]. We hypothesize that signals generated by streptomycin converge on *comCDE* through several pathways. Indeed, aminoglycosides are known to generate HS-like damage [56], while also triggering hydroxyl radical formation [63,64] and perturbing DNA replication [65].

Finally, these observations suggest that *comCDE* is the focal point of distinct stress signalling pathways, supporting the view that pneumococcal competence is a general stress response.

## Materials and methods

### Strains and growth medium

The pneumococcal strains, primers, and plasmids used in this study are listed in S1 Table and S2 Table.

Certain strains were rendered unable to spontaneously develop competence either by deletion of the *comC* gene (*comC0*) [2] or by deletion of *comA*, the CSP exporter [66]. Previously published constructs and mutants were transferred from published strains by transformation with appropriate selection (S1 Table).

Site-directed mutagenesis was based on strand overlap extension (SOE) [67]. Briefly, two PCR reactions, with primer pairs A1-A2 and B2-B1, were used to generate fragments A and B that incorporate a mutant primer (A2) at one extremity of A and its complement (B2) at the other, overlapping, extremity of B. A third PCR reaction with primer pair A1-B1 then produces a unique fragment with the mutant sequence in the middle (S2 Table). The resulting fragment was used to transform a recipient strain as described below. Potential transformants, typically 10–20, were sequenced to identify the new mutant isolate (Eurofins genomics).

*htrA* and *parB* gene mutants were generated by *in vitro* Mariner insertion as previously described [68] (see S1 Table and S2 Table).

Standard procedures for transformation and growth media were used [69]. Briefly, pre-competent cells are treated at 37˚C for 10 min with synthetic CSP1 (100ng mL$^{-1}$) to induce competence, then exposed to transforming DNA for 20 min at 30˚C. Transformants were then plated on CAT agar supplemented with 4% horse blood followed by incubation for 120min at 37˚C. Transformants were then selected by addition of a second layer of agar medium containing the appropriate antibiotic and overnight incubation at 37˚C. Antibiotic concentrations (μg mL$^{-1}$) used for the selection were: chloramphenicol (Cm), 4.5; kanamycin (Kan), 250; spectinomycin (Spc), 100; streptomycin (Sm), 200 and erythromycin (E), 0.1. Unless otherwise described, pre-competent cultures were prepared by growing cells to an OD$_{550nm}$ of 0.1 in C +Y medium (pH 6.8) before 10-fold concentration in C+Y medium supplemented with 15% glycerol and storage at –80˚C.

## Monitoring of growth and luciferase expression

For monitoring of competence gene expression using transcriptional fusions, an *S. pneumoniae* DNA promoter fragment (*Hin*dIII—*Bam*HI) was inserted upstream of the *luc* gene carried by pR424 [24] or pR414 plasmid [68]. Homology-dependent integration of the non-replicative recombinant plasmid into the pneumococcal chromosome was selected using chloramphenicol or erythromycin resistance respectively.

For the monitoring of growth and luciferase expression, precultures were gently thawed and aliquots were inoculated (OD$_{550nm}$ of 0.005) in luciferin-containing C+Y medium (pH 7.9) as previously described [69] and distributed into a 96-well (300 μl per well) white microplate with clear bottom (Corning). Relative luminescence units (RLU) and OD$_{492nm}$ values were recorded at defined time points throughout incubation at 37˚C in a Varioskan luminometer (ThermoFisher).

Transformation frequency was measured by mixing cells with a DNA fragment carrying the point mutation rpsL41 that confers streptomycin resistance. The 3434pb *rpsL* PCR fragment was generated using primers MB117 (AATCTCCGCTGTAGGTCACTTTCTT) and MB120 (TTGGATTGGGTGTGCATTTGC). After 300 minutes, cells were collected and plated on selective and non-selective media.

## Heat shock and monitoring of growth and luciferase expression

Precultures of *comA*$^-$ or *comC$_0$* cells were gently thawed and aliquots were inoculated (OD$_{550}$nm of 0.005) in luciferin-containing C+Y medium (pH 7.9) as previously described [70] and grown for 30 minutes at 37˚C. Cultures were then split into equal volumes (700μl) and grown for a defined period (usually 15 minutes) at the desired temperature (generally 37 or 42˚C). Growth and luciferase expression were monitored as above. Where appropriate, fold induction (FI) of the expression of the various genes of interest was calculated as the ratio of the area under the curve of heat-exposed cells to the area under the curve of unexposed cells. The areas under the curve were estimated by summing the values of the luciferase activity values (RLU/OD) over the first 60 minutes of the experiment. For antibiotics exposure, fold induction (FI) is calculated over the first 200 minutes of the experiment. The means of the ratio and standard deviation are the averages of at least three independent determinations made on different days.

## Antibiotic exposure and monitoring of growth and luciferase expression

Precultures of *comA*$^-$ or *comC0* cells were gently thawed and aliquots were inoculated (OD$_{550nm}$ of 0.005) in luciferin-containing C+Y medium (pH 7.9) as previously described [70] and grown for 30 minutes at 37˚C. Cultures were then split into equal volumes (700μl) and

grown for a defined period (usually 15 minutes) with or without antibiotics; norflaxacin (1.25 μg mL$^{-1}$) or streptomycin (6.25 μg mL$^{-1}$). Following incubation, cells were centrifuged (5 min at 5 000 rpm) and then washed with and finally resuspended in fresh C+Y medium (pH 7.9). Growth and luciferase expression were then monitored as above. Fold induction (FI) was calculated as for the heat shock procedure but the luciferase activity values were summed over the first 200 minutes of the experiment.

### Fluorescence microscopy and image analysis

Pneumococcal cultures were grown in 0.7 mL C+Y medium from OD$_{550nm}$ 0.004 to 0.1. Cells were then incubated at 42˚C or 37˚C for 15 minutes. The heat-exposed cells and unexposed controls were incubated at 37˚C for 20 minutes. In parallel, competence was induced in one culture by addition of CSP 200 ng mL$^{-1}$ for 15 minutes before visualisation. 1μL of each culture was spotted on an agar pad of 1.2% agarose in C+Y medium on a microscope slide, as previously described [71] and observed using a Nikon ECLIPSE Ti microscope. Images were captured and analysed with NIS-element AR software (Nikon).

For image analysis, cells were detected with the threshold command of NIS-element software. The GFP fluorescence of each cell was then measured with the automatic measurement option of mean FITC. For each image, the background fluorescence of the agar slide was measured from the average fluorescence of 6 different cell-free regions of interest. This average background was subtracted from the fluorescence measured for each cell of the image. Violin plot representations and statistical analyses (Mann-Whitney test) were performed with Rstudio software.

### Western blot analysis

Sample preparation was as for fluorescence microscopy experiments. After growth, 1mL of each culture was centrifuged at 5,000 rpm for 5 minutes. Pellets were resuspended in 40 μL of SEDS (0.15 M NaCl, 0.015 M EDTA, 0.02% SDS and 0.01% sodium deoxycholate) and incubated for 10 minutes at 37˚C to promote cell lysis. 5X sample buffer containing 10% β-mercaptoethanol was added and samples were incubated 10 minutes at 90˚C and stored at -20˚C until required. Samples were then normalized according to the initial OD$_{550nm}$ reading, and loaded onto SDS-PAGE gels 4–12% Bis-Tris (Invitrogen). After electrophoresis for 50 min at 200V, proteins were transferred onto a nitrocellulose membrane using a Transblot Turbo (BIO-RAD) apparatus. Membranes were incubated for 1 hour at room temperature in 1x TBS with 0.1% Tween20 and 10% milk, before being washed twice in 1x TBS with 0.1% Tween20 and probed with primary antibodies (1/10,000) in 1x TBS with 0.1% Tween20 and 5% milk overnight at 4˚C. After a further four washes in 1x TBS with 0.1% Tween20, membranes were incubated with anti-rabbit secondary antibody (1/10,000) for 90 min, then washed four times in 1x TBS with 0.1% Tween20. Clarity Western ECL Substrate (BIO-RAD) was used to detect the proteins by chemiluminescence using Fujifilm LAAS 4000 and software Image Reader (Fuji).

### Supporting information

**S1 Fig. Heat exposure promotes transformation.** A. Comparison of transformation efficiencies in strain R825 growing in C+Y medium at 37˚C after 15 minutes of exposure to 37, 39, 42 or 45˚C. Saturating (100 μg mL$^{-1}$) concentrations of rpsL41 PCR fragment, conferring streptomycin resistance via point mutation, was used as transforming DNA. B. As in A, but for various periods at 42˚C only. C. As in A, but in strain R895 (*ssb*::*luc*) after 15 minutes at 42˚C only.

Data represented as Mean ± standard deviation of triplicate repeats.
(TIF)

**S2 Fig. The availability of the CSP is a limiting factor for fully triggering the competence.**
A. As in Fig 2A for strain R825 exposed for 15 minutes to 42˚C (red lines) or 37˚C (blue lines). CSP (100 ng mL$^{-1}$) was added 20 minutes after first measurement (open triangles) or not (solid circles). For clarity, only a single data set, representative of three independent determinations made on different days, is presented. B. Enlargement of the early post-heat growth phase from panel A. Red; 15 minutes exposure to 42˚C. Blue; 15 minutes exposure to 37˚C.
(TIF)

**S3 Fig. Heat-induced *comCDE* expression is specific to the P$_E$ class promoter.** A. Schematic representation of the *comCDE* locus and upstream region. Promoters driving the basal level of *comCDE* transcription are represented by bent arrows. Straight arrows represent transcription activity; the dashed arrow represents transcription bypassing the terminator terT. B-D. Growth of pneumococcal cells in Fig 3B–3D respectively strains R1521 (*comC*::*luc*), R1694 (*tRNA$^{arg5}$*::*luc*) and R4639 (*amiA*::*luc*). Red; 15 minutes exposure to 42˚C. Blue; 15 minutes exposure to 37˚C.
(TIF)

**S4 Fig. The burst after heat exposure is specific to early competence genes.** Growth of pneumococcal cells in Fig 4A–4G: A. R1521 (*comC*::*luc*); B. R1548 (*comA*::*luc*); C. R2200 (*comX2*::*luc*); D. R2218 (*comX1*::luc); E. R3688 (*comW*::*luc*); F. R1502 (*ssbB*::*luc*), G. R2448 (*dprA*::*luc*). Red; 15 minutes exposure to 42˚C. Blue; 15 minutes exposure to 37˚C.
(TIF)

**S5 Fig. The *comCDE* burst requires a functional ComD/E two component system.** Growth of pneumococcal cells in Fig 5A–5F. A. R1521 (*comC*::*luc*); B. R1627 (*comC*::*luc*, *comE*$^-$); C. 4585 (*comC*::*luc*, *hexA*$^-$); D. R1798 (*comC*::*luc*, *hexA*$^-$, *comE$^{D58A}$*); E. R1628 (*comC*::*luc*, *comD*$^+$), F. R1648 (*comC*::*luc*, *comD*$^-$). Red; 15 minutes exposure to 42˚C. Blue; 15 minutes exposure to 37˚C.
(TIF)

**S6 Fig. HtrA is required to conduct the heat exposure signal to *comCDE*.** Growth of pneumococcal cells in Fig 6A–6D. A. R1521 (*comC*::*luc*); B. Strain R2017 (*comC*::*luc*, *dprA*$^-$); C. R1526 (*comC*::*luc*, *clpP*$^-$); D. Strain R4578 (*comC*::*luc*, *htrA*$^-$). Red; 15 minutes exposure to 42˚C. Blue; 15 minutes exposure to 37˚C. E. Fluorescence in cells of strains R4254 and R5129 (*htrA*$^-$) carrying the *gfp* gene under the control of a ComE-dependent promoter. F. Violin plots representing mean GFP fluorescence intensity in non-HS (blue) and HS (red) cells. Boxes extend from the 25th percentile to the 75th percentile, with the horizontal line at the median. Dots represent outliers. n = number of cells analysed for strains R4254 and R5129 in the different conditions. *** = p-value < 0.001.
(TIF)

**S7 Fig. Catalytic and activation residues of HtrA protease are required to activate *comCDE* transcription.** A. Schematic representation of the *htrA*/*parB* operon and position of relevant residues of HtrA. B-H. Growth of pneumococcal cells in Fig 7B–7H: B. R875 (*comC*::*luc*); C. R4657 (*comC*::*luc*, *parB*$^-$); D. R4629 (*comC*::*luc*, *htrA$^{F11stop}$*); E. R4630 (*comC*::*luc*, *htrA$^{S234A}$*); F. R4676 (*comC*::*luc*, *htrA$^{Q224A}$*); G. R4684 (*comC*::*luc*, *htrA$^{R206A}$*); H. *htrA* expression in strain R2813 (*htrA*::*luc*). Red; 15 minutes exposure to 42˚C. Blue; 15 minutes exposure to 37˚C.
(TIF)

**S8 Fig. Several molecular pathways convey different signals to modulate *comCDE* basal level transcription.** A-D. Growth of pneumococcal cells growing in C+Y medium at 37˚C after 15 minutes exposure to Norfloxacin (1.25 μg mL$^{-1}$) (red symbols) or no antibiotic (blue symbols) prior to first measurement. A. R875 (*comC::luc*); B. R4629 (*comC::luc*, *htrA$^-$*); C. R4642 (*tRNA$^{arg5}$::luc*) D. R4641 (*amiA::luc*). E. To illustrate reproducibility, the results from Fig 8 panel A and Fig 8 panel B are presented in the form of individual experiments carried out separately. *comCDE* expression shown as relative luminescence unit per OD (RLU/OD) in strains growing in C+Y medium at 37˚C after 15 minutes exposure to Norfloxacin (1.25 μg mL$^{-1}$) (red symbols) or no antibiotic (blue symbols) prior to first measurement (solid line). Dashed lines correspond to respective growth curves derived from OD$_{492nm}$ measurements. Upper panels, R875 (parental strain). Lower panels, R4629 (*htrA$^-$*).
(TIF)

**S9 Fig. Streptomycin exposure promotes a partially HtrA dependent burst of transcription of *comCDE*.** A. *comCDE* expression shown as relative luminescence unit per OD (RLU/OD) in strains growing in C+Y medium at 37˚C after 15 minutes exposure to streptomycin (6.25 μg mL$^{-1}$) (red symbols) or no antibiotic (blue symbols) prior to first measurement (solid line). Dashed lines correspond to respective growth curves derived from OD$_{492nm}$ measurements. Upper panels, R875 (parental strain). Lower panels, R4629 (*htrA$^-$*). B. Same as A but for *amiA* expression (R4641). To illustrate reproducibility, the results are presented in the form of individual experiments carried out separately. Global fold induction (FI) for these three experiments is 1.55 +/-0.09 for strain R875, 1.03+/-0.05 for strain R4629 and 0.68+/-0.03 for strain R4641 (see materials and methods).
(TIF)

**S1 Table. Strains used in this study.**
(PDF)

**S2 Table. Oligonucleotides used in this study.**
(PDF)

## Acknowledgments

We thank Isabelle Mortier-Barriere for the support with fluorescence microscopy. We thank Calum Johnston and Nathalie Campo for critical reading and helpful editing of the manuscript. This work was supported by the Centre National de la Recherche Scientifique and University Paul Sabatier-Toulouse III.

## Author Contributions

**Conceptualization:** Mickaël Maziero, Patrice Polard, Mathieu Bergé.

**Formal analysis:** Mickaël Maziero, Patrice Polard, Mathieu Bergé.

**Funding acquisition:** Patrice Polard.

**Investigation:** Mickaël Maziero, Mathieu Bergé.

**Supervision:** Patrice Polard, Mathieu Bergé.

**Writing – original draft:** Mathieu Bergé.

**Writing – review & editing:** Mickaël Maziero, David Lane, Patrice Polard, Mathieu Bergé.

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
