## [Decision Letter · Decision Letter 0]

26 May 2023

Dear Dr Berge,

Thank you very much for submitting your Research Article entitled 'Fever-like temperature bursts promote competence development via an HtrA-dependent pathway in Streptococcus pneumoniae.' to PLOS Genetics.

The manuscript was fully evaluated at the editorial level and by three independent peer reviewers. The reviewers appreciated the attention to an important problem, but raised some substantial concerns about the current manuscript. Based on the reviews, we will not be able to accept this version of the manuscript, but we would be willing to review a much-revised version.   We cannot, of course, promise publication at that time.

Should you decide to revise the manuscript for further consideration here, your revisions should address each of the specific points made by each reviewer. It will be of particular importance to adress the reviewers' concers on the significance of the initial expression burst on transformation rates, confirm the htrA-effect in wild-type backgrounds and show how removal of htrA affects the growth. Information about the number of replicates that have been done to support the presetned data should be included. We will also require a detailed list of your responses to the review comments and a description of the changes you have made in the manuscript.

If you decide to revise the manuscript for further consideration at PLOS Genetics, please aim to resubmit within the next 60 days, unless it will take extra time to address the concerns of the reviewers, in which case we would appreciate an expected resubmission date by email to plosgenetics@plos.org.

We are sorry that we cannot be more positive about your manuscript at this stage. Please do not hesitate to contact us if you have any concerns or questions.

Yours sincerely,

Morten Kjos

Guest Editor

PLOS Genetics

Lotte Søgaard-Andersen

Section Editor

PLOS Genetics

Reviewer's Responses to Questions

**Comments to the Authors:**

Reviewer #1: In the manuscript titled, “Fever-like temperature bursts promote competence development via an HtrA-dependent pathway in Streptococcus pneumoniae”, Maziero et al demonstrate that transient heat shock results in two effects on the competence pathway - an immediate CSP-independent, small increase in basal expression of early competence genes as well as a later, CSP-dependent induction of the entire competence pathway. The authors focus on the earlier, CSP-independent effect in their work. Previous work had shown that increases in comCDE basal expression were dependent on the ComDE TCS. In agreement with that work, authors show that heat-shock dependent increase in comCDE basal expression was indeed dependent on ComDE signaling. This increase in the basal expression of comCDE was HtrA-dependent. To test if this initial burst of ComCDE expression was a general stress response, authors tested if brief exposure to antibiotics (norfloxacin and streptomycin) also had a similar effect. While exposure with norfloxacin led to this small increase in comCDE basal expression, albeit in an HtrA-independent way; the effect of streptomycin was suspect. The authors investigate an interesting topic that is of high importance for the field. Their findings show potentially interesting results, however, major gaps remain in mechanistic understanding of the pathways that authors attempt to explore in this work (as below).

Major Comments –

1. The authors mostly focus on the immediate increase in basal expression of ComCDE that fails to result in an induction of the late competence genes (Fig 4). In the absence of this induction, it is unclear what is the physiological significance of this small induction in ComCDE expression.

2. Fig 1 – It remains unclear why basal increases in ComCDE levels observed immediately following heat shock, which results in increased ComC levels, not result in a positive feedback loop that activates the competence pathway? Is CSP not being secreted? Is it being degraded?

3. It remains unclear how HtrA stimulates heat-shock induced increases in basal expression of comCDE. Many questions remain on this front –

a. Previous work (that authors cite in lines 283-288) has shown that HtrA negatively regulates competence development by degrading CSP. Since this work focuses on the CSP-independent increase in basal expression of ComCDE, it is unclear how HtrA is leading to the opposite phenotype – that of promoting competence development. Is it indeed the protease activity of HtrA that is responsible for this phenotype, and is so, what is the substrate that HtrA is acting on to result in this increase?

b. How does HtrA impact the CSP-dependent induction of the entire competence pathway observed ~75 min post heat shock (phenotype described in Figs 1, 2B)? Is that also HtrA-dependent?

c. Lines 228-229: Authors state that HS could allosterically stimulate HtrA catalytic activity. However, to conclusively show that temperature regulates HtrA allosterically, reaction kinetics and Kd values will be needed.

4. Fig 8 and lines 237-243:

a. Why is the basal expression of ComCDE in the non-antibiotic treated samples (blue lines) about 1-2 log10-higher than the corresponding samples (37C) reported throughout the paper? Shouldn’t these be comparable?

b. Data of a single replicate presented in Fig 7B shows high fluctuation in ComCDE expression upon streptomycin treatment. Without more replicates, it is difficult to assess whether brief streptomycin exposure indeed results in increased basal expression of ComCDE. Is the increase in expression statistically significant? Moreover, the effect of streptomycin-treatment the authors are alluding to in this figure materializes ~80 mins after antibiotic exposure, unlike the immediate effect observed post-heat shock. Due to lack of additional data, it seems plausible that this is a distinct mechanism at play.

Minor Comments –

1. Lines 192 and Fig 5A – Isn’t the burst of transcription observed in a ComC- background as opposed to WT?

2. Mismatches in the strain numbers in the figure legends and the strain table. For instance, for Fig 5C, strain 4585 not listed in the strain table; fig 5E, Is R1648 ComD+ strain (as in legend) or ComE+ as in strain table. Please re-check throughout the manuscript.

3. Lines 209-210 – Rephrase to… clpP mutants “do not abolish”…., since there seems to be a further increase in the ComCDE levels in the ClpP- strain.

4. What is the difference between Figs 7B and 7H? The figure legend suggests it is the same strain. Further, I could not find a reference to Fig 7H in the text.

5. Fig 8: Correct X-axis to time after “antibiotic” exposure.

Reviewer #2: The manuscript by Maziero et al. reports a novel signaling pathway that allows Streptococcus pneumoniae to activate competence by sensing a temperature upshift (similar to fever). The authors show that this physical stress is perceived by the communication module ComCDE through activation of the chaperone/protease HtrA. This priming of ComCDE is than amplified later through the positive feedback loop via the pheromone CSP, resulting in the massive activation of the late competence phase. Nevertheless, the molecular mechanism (direct or indirect) ensuring the relay between HtrA activation and the ComCDE system remains to be elucidated. In addition, the authors show that this novel pathway of competence activation in the pneumococcus is not used by antibiotics targeting DNA replication and partially used by antibiotics targeting the ribosome. This work is of importance since the signaling pathways sensing physicochemical stresses resulting in competence development remain largely a black box in streptococci.

Globally, the manuscript is well-organized with convincing genetic experiments that demonstrate the relay between a temperature upshift, HtrA activation and ComCDE activation. The authors should carefully edit their manuscript for typographical errors (a non-exhaustive list is available in minor comments).

Major comments:

1/ The authors show that the late competence phase could be activated by a temperature upshift in pH conditions that do not lead to spontaneous transformation. However, I think that its important to show that transformation events are really taking place. In addition, it would be interesting to show that the transformation rate is responding to the increase in temperature or duration of the upshift.

2/ Although the relay between activated HtrA and ComCDE could be complex, a simple model where activated HtrA directly interacts in the membrane with the histidine kinase ComD to modulate the phosphorylation level of ComE could be easily tested either using a bacterial two-hybrid system or Split-nanoluc (functional in the pneumococcus). Although not fully required, such experiments could reinforce the story and give additional novelty.

Minor comments:

L18-L19: “in the physical conditions of the environment, oxidative potential and temperature”. I do not see the point here to cite oxidative potential. I would remove “oxidative potential and temperature”. Change in “in physical conditions of environment”

L22: “We find that certain other stimuli”. This is unprecise and needs to be rephrase.

L24: “by any of several independent pathways”. Again, lack of precision

L39: “is essential to relaying of the thermal signal” -> is essential for relaying the thermal signal

L84-85: “such has that of Lee” -> such as. “That proteins” -> those proteins

L85-86: rephrase the sentence -> “, suggesting cross regulation

L108: give the initial pH value here

L142: “with a comC-minus strain” -> the minus superscript is not clearly visible

L190: “luciferase” -> remove italics

L224: “two key residues” -> cite those two residues here

L343: remove “, this”

L398-399: remove “was calculated as follows”

L424: performed with

L625-628: Fig 2A and B. Please indicate how many independent repetitions were performed

Reviewer #3: In this manuscript the authors study the role of thermal stress on

competence development in the pneumococcus. To do this, they track the

expression of several competence genes after a short exposure to high

temperature and find that heat shock induces a burst in expression of

the early competence genes that is mediated by ComDE. Using a set of

deletion mutants they explore which other genes might mediate this

induction and identify htrA as a potential candidate for transmitting

the heat shock signal to the competence operons. Although previous

studies had already established a link between temperature and

competence regulation (e.g. Luo et al 2003), this manuscript

systematically quantifies this effect across different competence

genes and sheds light on a potential mechanism by identifying that

induction of the com genes occurs through the PE promoter. While the

evidence that they present in this respect is convincing, I find that

the data on the involvement of htrA is less solid. In fact, I was

surprised by their findings regarding HtrA given that previous work

has shown that this protein degrades CSP and potentially ComD and thus

might inhibit rather than induce competence development. The main

source of evidence for the presence of an HtrA-dependent pathway is

that the initial burst in comCDE expression is not observed in htrA

deletion mutants after heat shock. However, since the authors do not

present the OD data it is unclear whether the htrA deletion mutants

have a growth defect during the first 100min (which in fact wouldn’t

be surprising given that previous studies have identified such growth

defects upon thermal stress when htrA is deleted). Moreover, this

experiment was done using a comC deletion background which prevents

the appearance of the second large competence peak that the authors

observe when the wildtype is exposed to thermal stress (Figure 1A). In

this sense, I could imagine that an htrA deletion mutant in a WT

background upon exposure to 42C could look like the WT exposed at 45C

(Fig 1A). In this case, the initial burst of comC expression is not

observed because of the long lag but the second competence peak

appears once the culture grows. The authors should construct the htrA

deletion mutant in a WT background and show that competence is never

induced (as it happens when the cultures are not exposed to heat) to

confidently claim that HtrA is necessary for the heat shock signal to

be transmitted to the com operon. Moreover, to further disentangle the

effect of deleting htrA on growth upon heat stress and competence

induction I’d suggest that the authors perform the single cell

competence assays done in Fig 2 with the htrA deletion mutants and

show that the fluorescence distributions are similar in heat exposed

and unexposed cultures.

Also, the fact that htrA can ‘transmit’ antibiotic stress upon

exposure to Streptomycin but not Norfloxacin could be constant with

the explanation above: HtrA might be necessary to prevent growth

defects upon exposure to Streptomycin given that this antibiotic

interferes with protein synthesis unlike Norfloxacin which targets DNA

replication. Thus, deleting htrA might affect growth upon exposure to

Streptomycin which can in turn translate in the absence of an initial

burst of competence induction not because HtrA is directly interfering

and transmitting signals to the comCDE operon but rather because of

the general physiological state of the cells. These experiments were

performed in the comA deletion background so as before I’d suggest

that the authors repeat this with htrA deletion mutants in a WT

background and assess competence development through the full growth

curve.

Other comments:

-Line 151. Unclear phrasing.

- Line 156. To streamline the story I’d suggest that the data showing

that the PE promoter is involved in competence induction by heat shock

and the data showing that ComDE is necessary for the upregulation of

competence genes upon heat stress (now in Figures 3 and 5) are

presented together.

- The accompanying growth curves should be presented for all figures

either in the SI or in the main text (the authors could add another

axis to RLU/OD plots to have both pieces of the data in the same

figure instead of creating an additional panel).

- Use a log scale for the RLU/OD values. In this way you can also use

the same axis range for all plots which would be especially useful for

comparing panels in the same figure (e.g. Fig 2).

- Use consistent notation for gene deletions (e.g. Fig 6)

- Figure 7. What is the difference between panels A and H? and if

they’re different what’s the purpose of the data presented in H?

- Figure 8. Legend should read ’Time after antibiotic exposure’. Also,

shouldn’t the blue curves in panel A and B be the same in panels A and

B? There is a 10-fold difference in the RLU/OD signal and as far as I

understand these correspond to either the WT or the htrA deletion

mutant in the absence on antibiotic exposure and thus should be the

same conditions in panels A and B.

- Line 399. Is this done with the RLU or RLU/OD values? Please give

more details on how the calculation is done.

**Have all data underlying the figures and results presented in the manuscript been provided?**

Reviewer #1: **No: **Authors only show data for a single replicate throughout the manuscript.

Reviewer #2: Yes

Reviewer #3: **No: **

PLOS authors have the option to publish the peer review history of their article (what does this mean?). If published, this will include your full peer review and any attached files.

Reviewer #1: No

Reviewer #2: No

Reviewer #3: No

---

## [Decision Letter · Decision Letter 1]

7 Aug 2023

Dear Dr BERGE,

Thank you very much for submitting your Research Article entitled 'Fever-like temperature bursts promote competence development via an HtrA-dependent pathway in Streptococcus pneumoniae.' to PLOS Genetics.

The manuscript was fully evaluated at the editorial level and by independent peer reviewers. The reviewers appreciated your response to their questions on the previous version on the manuscript. One of the reviewers has some additional comments that we ask you to respond to. Comments number 1, 2, 5, 6 and 7 from the reviewer should be adressed in the revised manuscript.

In addition, I would suggest some minor corrections: line 108: remove the extra "C". Figure 6G: capital "A" in "htra".

We ask you to modify the manuscript according to the recommendations. 

Yours sincerely,

Morten Kjos

Guest Editor

PLOS Genetics

Lotte Søgaard-Andersen

Section Editor

PLOS Genetics

Reviewer's Responses to Questions

**Comments to the Authors:**

Reviewer #1: The authors have addressed my concerns.

Reviewer #3: I thank the authors for providing the additional growth data and performing the single-cell experiments in the htrA mutants. Some additional comments below:

1. After re-reading the abstract, I think the authors should remove the claim that 'there have been no reports of the response of the pneumococcal competence regulon to changes in the physical conditions of the environment'. The effect of environmental variables such as pH and environmental diffusivity has been thoroughly explored before (Yang et al 2010; Domenech et al 2020; Moreno-Gamez et al 2017) and as I mentioned in my previous review even the role of temperature is not something that the authors are establishing for the first time.

2. I found it surprising that the htrA mutant does not only lack a growth defect upon heat stress but it even seems to grow better than the WT (fig 6F vs 6H). I’d appreciate if the authors include a sentence in the discussion about this. I did a quick search and couldn’t find studies reporting similar observations upon deleting htrA - deletion mutants always seem to underperform under stressful conditions relative to the WT.

3. Regarding my suggestion about using a log scale I don’t understand what the authors mean with ‘… it graphically overwrites the initial burst of early gene transcription.’, especially since the log-scale would have precisely the opposite effect of amplifying that initial burst (e.g. in Figure 2A the two-fold increase in expression level that the authors observe would be ~1/4th of the second peak). Also, as I suggested the first time please use the same y-axis range in all panels in the same figure because otherwise it becomes hard to compare them.

4. Given the variability observed for Streptomycin it’d be great if the authors show additional replicates for the Norfloxacin data. It seems like now the authors are presenting data from only a single experiment.

5. Line 303. It’d be great if the authors could reflect on why their results are different from the ones obtained by Stevens et al besides just stating the observations are not consistent among both studies.

6. Line 354. Looking at figure S8 I’m not sure to which initial burst the authors are referring for the Streptomycin data. In fact, it looks like that first burst is reduced in the Streptomycin treatment relative to the control and only later the full peak appears. Please clarify this.

7. There is a typo in the y-axis labels of Figure S8.

**Have all data underlying the figures and results presented in the manuscript been provided?**

Reviewer #1: Yes

Reviewer #3: Yes

PLOS authors have the option to publish the peer review history of their article (what does this mean?). If published, this will include your full peer review and any attached files.

Reviewer #1: No

Reviewer #3: No

---

## [Editor Report · Decision Letter 2]

30 Aug 2023

Dear Dr BERGE,

We are pleased to inform you that your manuscript entitled "Fever-like temperature bursts promote competence development via an HtrA-dependent pathway in Streptococcus pneumoniae." has been editorially accepted for publication in PLOS Genetics. Congratulations!

Yours sincerely,

Morten Kjos

Guest Editor

PLOS Genetics

Lotte Søgaard-Andersen

Section Editor

PLOS Genetics

Comments from the reviewers (if applicable):

**Data Deposition**

http://datadryad.org/submit?journalID=pgenetics&manu=PGENETICS-D-23-00443R2

**Press Queries**

---

## [Editor Report · Acceptance letter]

7 Sep 2023

PGENETICS-D-23-00443R2 

Fever-like temperature bursts promote competence development via an HtrA-dependent pathway in Streptococcus pneumoniae. 

Dear Dr BERGE, 

We are pleased to inform you that your manuscript entitled "Fever-like temperature bursts promote competence development via an HtrA-dependent pathway in Streptococcus pneumoniae." has been formally accepted for publication in PLOS Genetics! Your manuscript is now with our production department and you will be notified of the publication date in due course.

With kind regards,

Zsofi Zombor

PLOS Genetics

On behalf of:
